



# The Ensemble Framework For Flash Flood Forecasting (EF5) v1.2: Description and Case Study

Zachary L. Flamig[1,2], Humberto Vergara[1,2], and Jonathan J. Gourley[2*]

[1]Cooperative Institute for Mesoscale Meteorological Studies, University of Oklahoma, 120 David L Boren Blvd, Norman, OK, USA 73072
[2]National Oceanic and Atmospheric Administration's National Severe Storms Laboratory, 120 David L Boren Blvd, Norman, OK, USA 73072

*Correspondence to:* Jonathan J. Gourley (jj.gourley@noaa.gov)

**Abstract.** The Ensemble Framework For Flash Flood Forecasting (EF5) was developed specifically for improving hydrologic predictions to aid in the issuance of flash flood warnings by the U.S. National Weather Service. EF5 features multiple water balance models and two routing schemes which can be used to generate ensemble forecasts of streamflow, streamflow normalized by upstream basin area (i.e., unit streamflow), and soil saturation. EF5 is designed to utilize high resolution precipitation

forcing datasets now available in near real time. A study on flash flood scale basins was conducted over the conterminous United States using gauged basins with catchment areas less than 1,000 km$^2$. The results of the study show that the three uncalibrated water balance models linked to kinematic wave routing are skillful in streamflow prediction.

## 1  Introduction

Flash floods are defined by an extreme flow into a normally dry area or a rapid water level rise above a threshold flood level. Typically, flash flood events begin within minutes to a few hours after the causative rainfall event although the timing can vary in different parts of the world  (NWS, 2016; WMO, 1988). An upper bound for the drainage area of basins is often considered as 1,000 km$^2$  (AMS, 2000). This definition of flash flooding also defines the requirements for any distributed hydrologic modeling system designed to forecast them. Such a system must be capable of cycling sub-hourly while providing forecasts

for at least 6 hours in the future. The system also must be able to resolve drainages with basin areas less than 1,000 km$^2$.

In the United States, floods and flash floods are the second deadliest weather phenomena behind heat  (Ashley and Ashley, 2008). Flash flood fatalities have previously been found to account for 80–90% of all flood fatalities. Globally, WMO (2008) found that there are currently 99 countries which issue flash flood warnings, but with 91 countries stating that further improvements to the warnings are necessary. The American Meteorological Society (AMS) policy statement on flash floods states,

"forecasting the time and location of flash floods requires high-resolution modeling of weather and water, assimilation of large data sets from high-resolution observations, and an integrated, coherent approach that allows meteorologists and hydrologists





to make rapid assessments and warning decisions" (AMS, 2017). Improved radar rainfall estimates are now available at 1 km$^2$ and 2 minute spatiotemporal resolution driving the demand for hydrologic forecasting systems which are capable of matching this resolution. This study will detail the development of a new high-resolution distributed hydrologic modeling framework which is capable of producing 0 to 24 hour forecasts of streamflow, unit streamflow, and soil saturation while ingesting high-

resolution radar rainfall estimates with a 10-minute update cycle. The goal of this framework is to be able to rapidly produce hydrologic assessments of flash flooding that guide operational warning decisions.

This study is part of the much larger Flooded Locations And Simulated Hydrographs (FLASH) project, which aims to provide NWS forecasters with better warning decision-support tools for issuing flash flood warnings in the United States (Gourley et al., 2017). Specifically the goal of the project is to improve the spatial specificity, timing, and accuracy of flash

flood warnings by leveraging Multi-Radar Multi-Sensor (MRMS) rainfall products for high-resolution forward hydrologic simulation. This study documents the hydrologic models used for FLASH, their setup, and their performance over the current period of record for the available high-resolution precipitation forcing. These hydrologic models have already been used for experimental evaluations with National Weather Service (NWS) forecasters in the Hydrometeorological Testbed (HMT-Hydro) (Martinaitis et al., 2017) and the Flash Flood and Intense Rainfall (FFaIR) experiments (Barthold et al., 2015). In both

experiments, the hydrologic products presented here received favorable reviews. These hydrologic products have been used for experiments with automation in the warning-decision process by recommending locations for possible flash flood warnings (Argyle et al., 2017). This paper will provide a review of existing hydrologic models, document the hydrologic models used in EF5, and demonstrate the performance of the hydrologic simulations with a multi-year case study over the United States.

### 1.1 Review of Existing Hydrologic Models

Resolving extreme rainfall and flash flood events requires radar rainfall estimates coupled with distributed hydrologic models that need to be run at fine spatial resolution on the order of 100 m to 2 km with a temporal step that is sub-hourly (Rafieeinasab et al., 2015). With this requirement, several distributed hydrologic models were evaluated for their potential to be run in this fashion to capture flash flood events over the conterminous United States (CONUS). Given the focus on extreme rainfall events where contributions of surface fluxes into the atmosphere are small compared to the magnitude of the rainfall, it is sufficient

to examine models with one-way coupling of rainfall onto the land surface. The TREX distributed hydrologic model was one option, however the model attempts to be fully physical meaning that it requires very fine spatial resolution and time steps on the order of seconds in order to properly solve the equations (Velleux et al., 2008). Running it over the CONUS would require computational resources unavailable at the present time for flash flood forecasting. Since a fully physically based distributed hydrologic model is too computationally complex to run with the required cycling times there is a need to identify the trade off

required to run conceptual-based hydrologic models. A brief literature review follows to answer the question of how accurate are the physically based hydrologic models and can we produce equal forecasts and understanding with a conceptually simpler model?

Devia et al. (2015) provides an overview of the differences between empirical (statistical), conceptual (parametric), and fully physical hydrologic models. The authors provide valuable dialog recognizing that each formulation of a hydrologic model has





strengths and weaknesses and there is no one answer for the entire problem domain in hydrology right now. Empirical models are considered to be useful only for the specific watershed they are developed on and cannot be trivially extended into new watersheds. Empirical models also perform poorly for extreme events that occur outside of their training datasets. Conceptual models are defined as simple and easily to implement in software but require large amounts of data for calibration. Physically

based models require extensive amounts of data on processes often not observed by current sensor networks and suffer from an inability to scale to large collections of watersheds. They further state that, "Each model has various drawbacks like lack of user friendliness, large data requirements, absence of clear statements of their limitations etc. In order to overcome these defects, it is necessary for the models to include rapid advances in remote sensing technologies, risk analysis, etc. By the application of new technologies, new distributed models can be developed for modelling gauged and ungauged basins." This belief is also

held by the authors of this study which leads to the creation of EF5.

Beven et al. (2014) addresses the ever increasing spatiotemporal resolutions of hydrologic models and particularly the land surface models coupled to atmospheric weather prediction models. They argue that there is a lack of information available to validate hypotheses made in hyper resolution models which may lead to mistaken beliefs about the processes. Information from hyper resolution models is often presented to stakeholders but without adequate quantification of the uncertainty leading

to precise but inaccurate forecasts. Further, the information is presented where only part of the model is hyper resolution and, for example, the precipitation forcing may not support the ability to resolve details at the resolutions being presented on maps. Kuczera et al. (2010) address the problem of uncertainty in the forcing information used for hydrologic models and model structural error. They argue that because of uncertainties in the forcing information, averaging methods applied to obtain it, and hydrologic model structural error, no conceptual model should be presented in a deterministic way. The argument about model

structural error suggests that future modeling systems should be able to account for these uncertainties with different model structures. Micovic and Quick (2009) look at the complexity of model representation needed as the temporal resolution of the hydrologic model decreases. So as simulations move from long-term climate simulations at a daily time step to simulations for individual days with extreme flood events is there a need for more hydrologic model complexity? The results from the study are only valid over a single watershed but suggest that important hydrologic processes for extreme flooding are different than

the processes yielding good prediction skill at long time ranges.

Given the evidence above the choice of a hydrologic model for CONUS-wide flash flood prediction seems to fall to multiple conceptual models which are computationally efficient. The CREST distributed hydrologic model developed by Wang et al. (2011) was picked for initial inclusion into the modeling framework because of its use previously at the global scale. The SAC-SMA model, in a distributed fashion similar to HL-RDHM, was also picked for inclusion in the framework because of its

existing operational use by the U.S. National Weather Service (Koren et al., 2004; Burnash, 1995). Existing implementations of both water balance schemes were tied to specific projects with details that precluded the easy use with forcing at a 1 km$^2$ 2 minute resolution necessitating new implementations in more flexible tools.





## 2 Ensemble Framework For Flash Flood Forecasting (EF5)

### 2.1 Introduction

The ideas behind EF5 were to incorporate the CREST water balance model, SAC-SMA water balance model, and then have the runoff output from either of those force a river routing scheme. Kinematic and linear reservior wave routing were the first river routing schemes implemented because of their overall computational efficiency. Applying EF5 in different locations made it apparent that there was a need for snow parameterization so the Snow-17 parametric temperature index snow model (Anderson, 1976) was added to EF5. Additionally it was identified that for some use cases calibration of the hydrologic models was desirable so the DREAM automatic calibration scheme (Vrugt et al., 2009) was incorporated into EF5. EF5 also has limited data assimilation capabilities supporting only direct insertion which can also be used as a boundary condition to model a smaller area of a large watershed (Houser et al., 2012). Figure 1 is the flow chart for EF5 showing the various modules and options that can be utilized for distributed hydrologic modeling with a focus on flash flooding.

To pick an area to model the basic files must first be provided which includes Digital Elevation Map (DEM), Flow Direction Map (FDM), and Flow Accumulation Map (FAM). EF5 is resolution independent and will work with any DEM resolution having been tested from 0.5 meter to 12 kilometer. The downstream point to model is then identified as a "gauge" which may or may not also correspond to an observation measurement location. Groups of gauges can be collected into a "basin" which is fundamentally just a collection of gauges one wishes to model on and not necessarily a collection of gauges in the same physical watershed. Parameters for the models are specified on a per gauge basis and then applied everywhere upstream of the gauge as a multiplier onto the distributed values until the next gauge if there is one. The parameters are specified either as a distributed grid and then a multiplier value or as a single value that is applied uniformly across the watershed.

EF5 is written in C++ and currently contains 20,388 lines of code while supporting Linux, Mac OS X and Windows operating systems. Linux and Mac OS X are supported via binaries run from the shell command prompt while Windows features a fully-fledged graphical user interface (GUI). The Windows GUI provides very similar visual feedback when compared to the Linux and Mac OS X versions but in an easier to work with package.

EF5 currently supports several different options for file formats and map projections. The preferred file format for use with EF5 is Geotiff, which has the distinct advantage of including native compression capabilities reducing file sizes greatly. ESRI Arc ASCII grids are also supported as input options for all gridded fields. For precipitation input, MRMS binary, TRMM TMPA 3B42 realtime binary are all supported input options. While GPM geotiffs can be trivially converted with GIS software such as GDAL to be compatible with EF5.

EF5 was created in a modular way to support multiple model physics and to do so implements virtual base classes for the snow melt, water balance, and routing physics. The water balance base class is detailed below, and thus it is possible for any water balance model that can conform to this specification to be implemented into EF5. EF5 provides two input forcing variables for the water balance component, precipitation and potential evapotranspiration. The output variables are a fast flow (typically surface) component, slow flow (typically subsurface) component, and a soil saturation value.



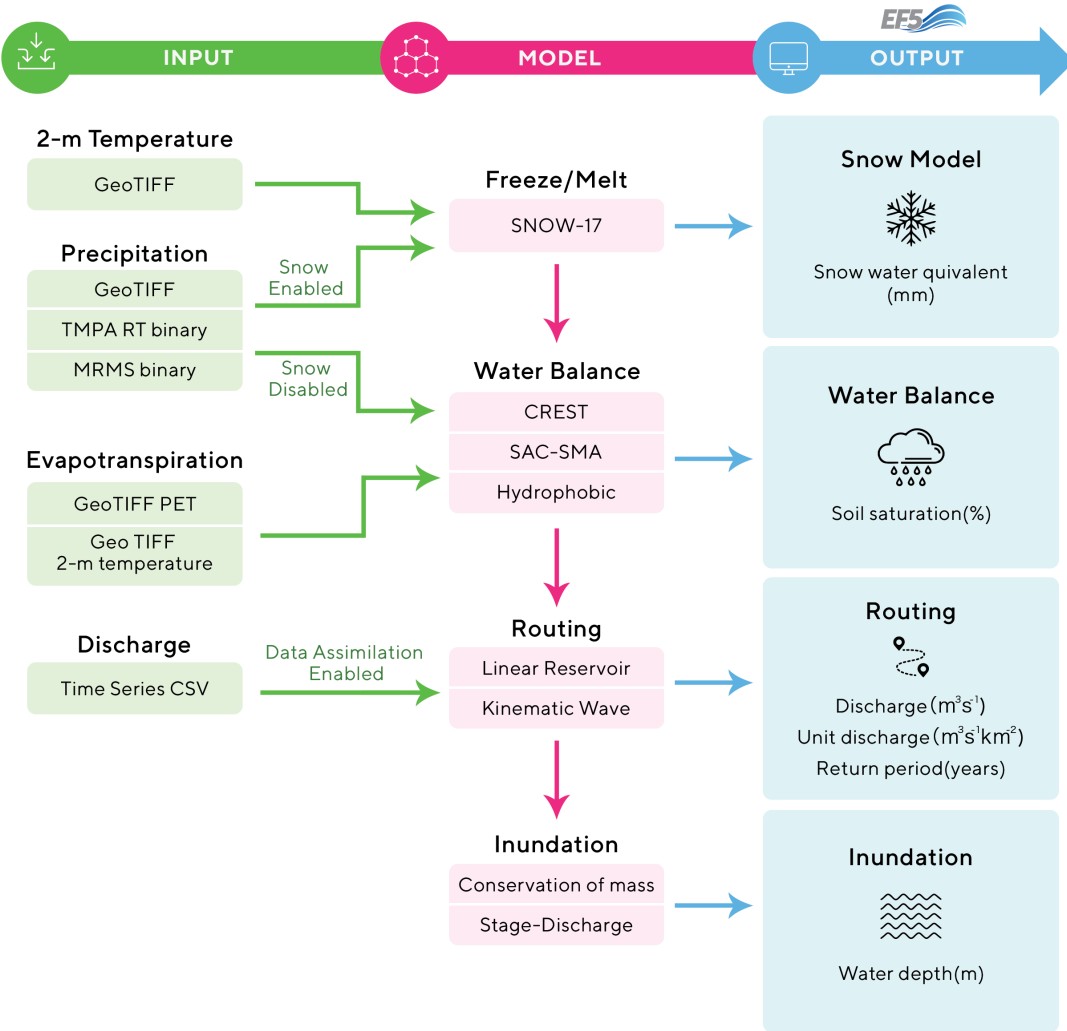

**Figure 1.** Flow chart illustrating the different modules and options available in EF5. The arrows show the paths available for input data to flow between the modules. The modules can be enabled or disabled from the control file to get the desired final model configuration.

The base class contains methods for initializing the model, initializing model state variables that may have been saved to file, saving model state variables to file and finally performing the water balance physics itself. For completeness the base classes for the routing and snow components are included below. The routing and snow components contain similar methods to be implemented as the water balance component with functionality for initialization, state loading and saving, and the main method for executing the physics. The routing virtual class takes a fast flow and slow flow input components and provides a single discharge output variable. The snow module takes as input precipitation and temperature while providing melted runoff (or just passing through precipitation in the no snow case) and snow water equivalent as the output variables.



This implementation of the model physics allows for EF5 to be easily expanded in the future to contain more options for treatment of basic hydrologic functions. This expandability is an important feature because it provides a way for new physics to be added to existing operational flood forecasting systems in the future without a complete overhaul of the supporting infrastructure.

## 2.2  Water Balance Models

Currently EF5 contains three water balance options. All three options are conceptually based and rely on parameters guided by land surface and subsurface properties measured in existing data sources. The three options described in this section are CREST, SAC-SMA and hydrophobic (HP). The most detailed description is provided for the CREST model because the underlying model has been modified from previous publications (Wang et al., 2011).

### 2.2.1  Hydrophobic (HP)

The HP option is by far the simplest, as there are no parameters to be specified for the land surface. The HP option treats the surface as completely impervious so all rain immediately runs off and flows downslope. The HP water balance option is included for the ability to diagnose processes and errors when running in an ensemble with the other water balance models. Underestimation of streamflow with the HP model indicates that the precipitation is likely biased. The HP model produces an upper bound on the expected discharge values. If the hydrophobic solution matches closely with the observed streamflow then either the entire drainage area is acting as an impervious surface or the inputs into the model are underestimating the magnitude of rainfall.

The hydrophobic model can also be used to approximate the land surface response after wildfires when the soils become hydrophobic. In watersheds mostly comprised of burn areas the HP option is expected to give the most realistic simulations. Running EF5 in an ensemble with all three water balance models allows for the impacts of wildfires to be accounted for without having to modify distributed model parameter grids. This allows for quicker operational response to changing land surface conditions in the event of a wildfire that is proceeded immediately by heavy rainfall events.

### 2.2.2  Coupled Routing and Excess Storage (CREST)

Another water balance option, CREST, is a derivation of the Xinanjiang model developed for use in China which features a variable infiltration curve for partitioning rainfall into direct runoff and infiltration (Ren-Jun, 1992; Liang et al., 1996; Liu et al., 2009). The first version of CREST was documented in (Wang et al., 2011) and the version used here is an adaptation of that. The EF5/CREST implementation has only a single soil layer, further simplifying the model and reducing the input data requirements. EF5/CREST also contains partitioning for impervious area. Figure 2 shows a schematic for the various processes represented in EF5/CREST to convert rainfall into runoff.

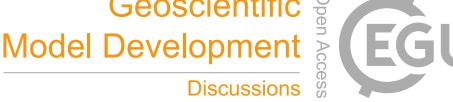



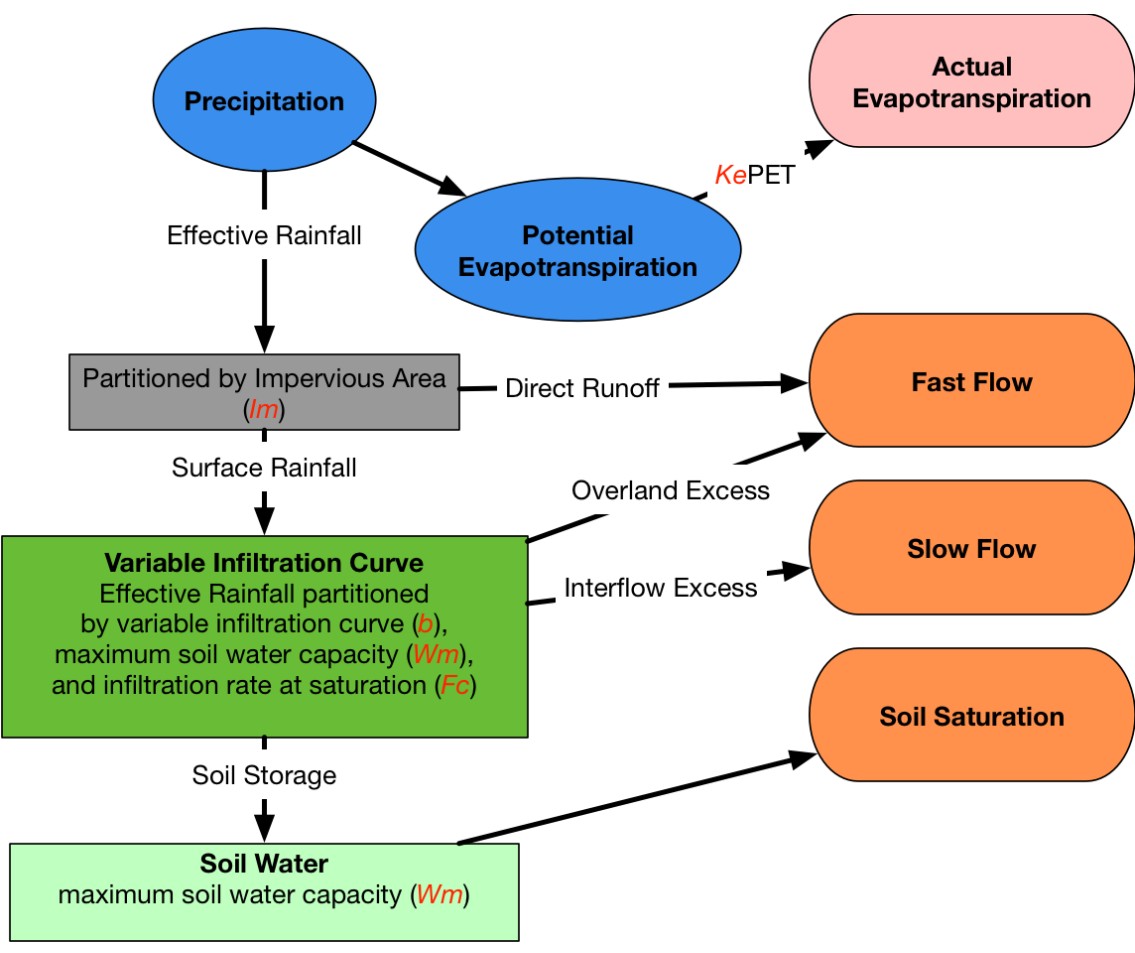

**Figure 2.** A schematic showing the progression of processes represented in the EF5/CREST water balance component.

Since EF5/CREST differs significantly from previous versions of CREST a detailed description of EF5/CREST is provided here. The first step is converting potential evapotranspiration to effective evapotranspiration using the configurable scalar parameter $K_e$ as shown in equation Equation 1. The $K_e$ parameter is typically set to 1.0 when working with distributed potential evapotranspiration and not utilizing model calibration.

$$5 \quad EET_t = K_e * PET_t \tag{1}$$

$PET_t$ is input forcing data into EF5 and $EET_t$ is the effective evapotranspiration. $PET_t$ in EF5/CREST is often computed using the Penman-Monteith equation (Montieth, 1965) which computes the potential evapotranspiration as a function of air temperature. Climatologies of air temperature can then be used to compute monthly mean or even hourly PET for use with





EF5.

$$EP_t = \begin{cases} 0, & \text{for } EET_t \geq P_t \\ P_t - EET_t, & \text{for } EET_t < P_t \end{cases} \tag{2}$$

$P_t$ is the input forcing rainfall into EF5. From the effective rainfall ($EP_t$) the direct runoff portion is calculated with the rest falling to the soil and then the infiltration process. The rainfall is then partitioned into a portion reaching the soil ($SP_t$), a

portion contributing to actual ET and a portion contributing to direct runoff ($DP_t$).

$$DP_t = EP_t * I_m \tag{3}$$

$$SP_t = EP_t * (1 - I_m) \tag{4}$$

$I_m$ is a scalar parameter representing the percent impervious area. One way the $I_m$ parameter is derived is using satellite-

based land use land cover (LULC) maps, which denote cities where land has been transformed into impermeable surfaces through human activity. The satellite LULC maps are typically at a very fine resolution which can then be averaged to the courser resolution of the model thus providing the percentage of impervious area per grid cell. The infiltration is then modeled using:

$$I_t = \begin{cases} 0, & \text{for } P_t \leq EET_t \vee SM_t \geq W_m \\ W_m - SM_t, & \text{for } (i_t + SP_t) \geq I_m \\ W_m - SM_t - W_m * [1 - \frac{i_t + SP_t}{i_m}]^{1+b}, & \text{for } (i_t + SP_t) < I_m \end{cases} \tag{5}$$

$W_m$ represents the maximum water capacity, $SM_t$ is the soil moisture state variable, and $b$ represents the exponent of the variable infiltration curve. Both $W_m$ and $b$ are parameters in EF5/CREST that are configurable but often defined apriori . $i_m$ represents the maximum infiltration capacity defined by:

$$i_m = W_m * (1 + b) \tag{6}$$

The infiltration capacity at the current time, $i_t$, is defined as:

$$i_t = i_m * [1 - (1 - \frac{SM_t}{W_m})^{\frac{1}{1+b}}] \tag{7}$$

The effective precipitation is then partitioned into excess rainfall ($ER_t$) based on the infiltration.

$$ER_t = \begin{cases} 0, & \text{for } SP_t = 0 \vee SP_t \leq I_t \\ SP_t - I_t, & \text{for } SP_t > I_t \end{cases} \tag{8}$$





The excess rainfall is then divided into overland ($OER_t$) and subsurface ($SER_t$) flow components by:

$$SER_t = \begin{cases} 0, & \text{for } EP_t = 0 \\ temX_t, & \text{for } ER_t > temX_t \\ ER_t, & \text{for } ER_t \leq temX_t \end{cases} \tag{9}$$

With $temX_t$ is defined as:

$$temX_t = \begin{cases} \frac{SM_t + W_t}{2W_m} * F_c, & \text{for } EP_t > 0 \\ (EET_t - P_t) * \frac{SM_t}{W_m}, & \text{for } EP_t = 0 \end{cases} \tag{10}$$

Using $F_c$ to represent the hydraulic conductivity and with $W_t$ as:

$$W_t = \begin{cases} 0, & \text{for } EP_t = 0 \\ W_m, & \text{for } SM_t + I_t \geq W_m \\ SM_t + I_t, & \text{for } SM_t + I_t < W_m \end{cases} \tag{11}$$

The overland flow component is then calculated by taking a difference between the amount that infiltrates and the excess rain plus adding in the direct runoff.

$$OER_t = \begin{cases} 0, & \text{for } EP_t = 0 \\ ER_t - SER_t + DP_t, & \text{for } EP_t > 0 \end{cases} \tag{12}$$

The new soil moisture value is then computed using:

$$SM_{t+1} = \begin{cases} SM_t - temX_t, & \text{for } EP_t = 0 \\ W_t, & \text{for } EP_t > 0 \end{cases} \tag{13}$$

Finally the actual evapotranspiration, $AET_t$, is given as:

$$AET_t = \begin{cases} temX_t & \text{for } EP_t = 0 \\ EET_t, & \text{for } EP_t > 0 \end{cases} \tag{14}$$

EF5/CREST has six configurable parameters. $W_m$ is the cell's maximum water capacity and is closely related to the soil

porosity over the first 50 to 100 cm of soil. This parameter controls how much water is necessary for a grid cell to become saturated and can be viewed as a bucket that fills up. $F_c$ is the maximum amount of water allowed to infiltrate into the subsurface flow when the grid cell is saturated. This parameter is closely related to saturated hydraulic conductivity. $K_e$ is a linear adjustment to potential evapotranspiration and controls how efficiently potential evapotranspiration is converted into actual evapotranspiration. The $b$ parameter is related to the soil texture. $I_m$ is the percent of rain that is converted directly into over-

land runoff. This parameter is related to the impervious area of the grid cell. The final parameter, $I_{wu}$ is the percent of $W_m$



that is water initially in the grid cell. This is really a model state, but to allow for more thorough model calibration is classed as a parameter value. Section 3.1 describes typical sources and gives examples of the EF5/CREST parameters described in this section.

### 2.2.3 Sacramento Soil Moisture Accounting (SAC-SMA)

5 The SAC-SMA water balance option is the most complex one featured in EF5 currently. The implementation of SAC-SMA in EF5 is based off the works of Koren et al. (2004) and Yilmaz et al. (2008) so the model structural details are not described here. Figure 3 is a schematic of the processes represented in the SAC-SMA water balance component. Multiple zones with significantly more complex interactions are included in SAC-SMA as compared with EF5/CREST. The twenty one parameters for EF5/SAC-SMA are listed and briefly described in Table 3. The SAC-SMA uses a saturation excess process to generate

10 runoff differing from the infiltration excess process used in EF5/CREST. Like EF5/CREST the Sacramento model utilizes a partition of rainfall between impervious and permeable surfaces with impervious area contributing directly to runoff in a grid cell.

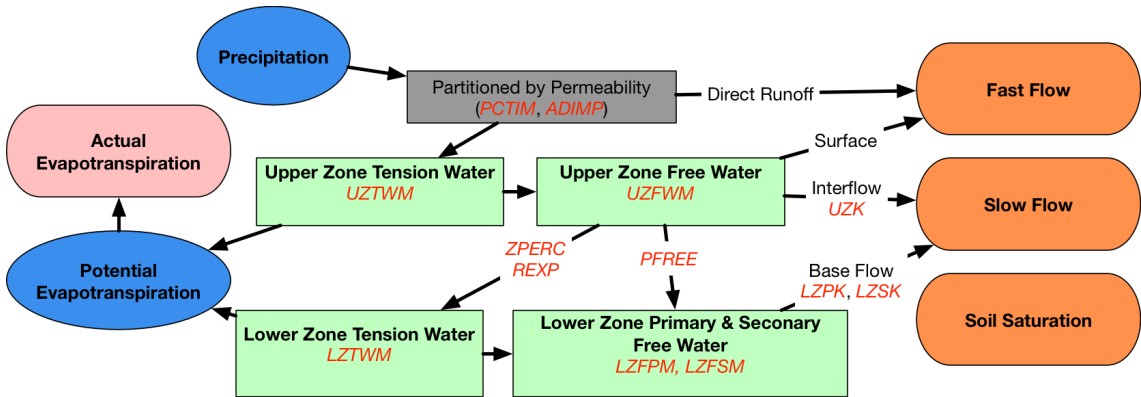

**Figure 3.** A schematic showing the progression of processes, inputs, and outputs represented in the EF5/SAC-SMA water balance component.

The EF5/SAC-SMA water balance model features an upper and lower layer (zone) which absorb and transmit water in conceptually different ways. The upper zone acts as the short term storage capacity for the grid cell so it is the first to fill when

15 rainfall occurs. The lower zone serves to provide the baseflow and acts as the long-term storage capacity for the grid cell. Each zone is further subdivided into tension water and free water. Tension water acts as surface tension and can only be removed from the grid cell by evapotranspiration. Free water can move through the cell vertically to the lower zone from the upper zone or discharged as streamflow out of the grid cell.



### 2.3 Routing Options

### 2.3.1 Linear Reservoir

The routing options available in EF5 are a lumped routing model conceptualized as a series of linear reservoirs and a kinematic wave approximation of the Saint-Venant equations for one-dimensional open channel flow. The linear reservoir option is

adapted from the original CREST model  (Wang et al., 2011) and has been well described and used in many hydrologic projects  (Nash, 1957; Moore, 1985; Chow et al., 1988; Vrugt et al., 2002). The EF5 linear reservoir option features two separate reservoirs where their depths are computed as:

$$OR_{t+1} = OR_t + OER_t + \sum_{i=1}^{N} OER_t^i \qquad (15)$$

$$SR_{t+1} = SR_t + SER_t + \sum_{i=1}^{N} SER_t^i \qquad (16)$$

where $OR_t$ and $SR_t$ are the overland and subsurface reservoirs. $OER_t$ and $SER_t$ are the excess rainfall components from EF5/CREST representing the fast and slow flow components. The $N$ represents the number of adjacent grid cells that flow into the current grid cell. The discharge out of each reservoir is based on the linear equations:

$$OQ_t = LeakO * OR_t \qquad (17)$$

$$SQ_t = LeakI * SR_t \qquad (18)$$

$$Q_t = OQ_t + SQ_t \qquad (19)$$

$LeakO$ and $LeakI$ are parameters defining the rate of discharge. The total discharge $Q_t$ is based on the summation of the fast

($OQ_t$) and slow ($SQ_t$) discharge rates. At each time step the fast and slow discharges are routed downstream following the FDM into the reservoir of the downstream grid cell.

### 2.3.2 Kinematic Wave

The implementation of the kinematic wave routing is based on an approximation to the one-dimensional unsteady open channel flow equations. The full one-dimensional unsteady open channel flow equations were developed in 1871 by Barré de Saint-

Venant and represent a physical description of the movement of water in a watershed  (Chow et al., 1988). The full equations have a number of assumptions that must be met including that the flow is one-dimensional, the flow varies gradually along the channel implying vertical accelerations can be neglected, the channel is approximately a straight line within a given grid





cell, the channel does not experience scour or deposition, and the flow fluid is incompressible implying a constant density. The kinematic wave model further simplifies the equations and requires that the bed slopes are steep. In the steep slope case the kinematic wave approximation reasonably describes the unsteady flow phenomena (Ponce, 1986). The work by (Ponce, 1991) claims that even in most overland cases the criterion for the kinematic wave approximation hold. The kinematic wave model

is widely used in hydrology and has been implemented in systems such as the Hydrologic Engineering Center's Hydrologic Modeling System (Feldman, 2000), the Storm Water Management Model created by the Environmental Protection Agency (Huber, 1995), HL-RDHM previously mentioned here and described in Koren et al. (2004), and finally already coupled to the Xinanjiang model (Liu et al., 2009).

Deriving the kinematic wave approximation starts with the Saint-Venant equations in the Eulerian frame of reference where

we model fluid as it passes by a control point, or in this case as it passes through a control volume. The time rate of change of the fluid is modelled as a function of the external forces acting on it as in Reynolds transport theorem (Chow et al., 1988). The external forces in this case are derived from Newton's second law of motion while neglecting lateral inflow, eddies and wind shear. The Saint-Venant continuity equation is given as:

$$\frac{\partial Q}{\partial x} + \frac{\partial A}{\partial t} = q \tag{20}$$

where $Q$ is the discharge, $x$ is the horizontal distance, $q$ is the lateral inflow into the channel, $t$ is time, and the channel cross-sectional area is $A$. The equation of momentum is defined by:

$$\frac{1}{A}\frac{\partial Q}{\partial t} + \frac{1}{A}\frac{\partial}{\partial x}\left(\frac{Q^2}{A}\right) + g\frac{\partial y}{\partial x} - gS_o + gS_f = 0 \tag{21}$$

where gravity is $g$, $S_o$ is the bottom channel slope, and $S_f$ is the friction slope. The terms in equation Equation 21 have been named such that $\frac{1}{A}\frac{\partial Q}{\partial t}$ is the local acceleration, $\frac{1}{A}\frac{\partial}{\partial x}\left(\frac{Q^2}{A}\right)$ is the convective acceleration, $g\frac{\partial y}{\partial x}$ is the pressure force, $gS_o$

is the gravity force, and $gS_f$ is the friction force. Simplifications to equation Equation 20 and Equation 21 represent different schemes commonly used in distributed hydrologic models. When no simplifications are made the routing is referred to as dynamic wave, when the acceleration terms are neglected the resulting wave model is called diffusive wave, and when the acceleration terms are neglected and the gravity force and friction force are assumed to be equal the result is the kinematic wave routing. In the kinematic wave assumption the resulting equation for momentum is:

$$Q = \alpha A^\beta \tag{22}$$

where $\alpha$ and $\beta$ are the kinematic wave parameters. This can be substituted back into the continuity equation and solved for Q which yields:

$$\frac{\partial Q}{\partial x} + \alpha\beta Q^{\beta-1}\frac{\partial Q}{\partial t} = q \tag{23}$$





Chow et al. (1988) also provides an implicit solution to the equations for distributed routing which is implemented in EF5. The kinematic wave routing in EF5 is applied only to the overland discharge, the subsurface discharge is routed with linear reservoir routing as described above. The equations above describe the kinematic wave routing for channel routing. For overland routing the process is the same as above but for $q$ instead of $Q$. The resulting equation is as follows:

$$\frac{\partial q}{\partial x} + \alpha_0 \beta_0 q^{\beta_0 - 1} \frac{\partial q}{\partial t} = i - f \tag{24}$$

where $\alpha_0$ is the overland conveyance parameter, and the $\beta_0$ parameter is fixed at $\frac{3}{5}$. The $i - f$ forcing term is the surface excess rainfall passed in from the water balance model. Table 4 details the parameter options for kinematic wave routing used by EF5.

## 3 Case Study Over the CONUS

### 3.1 EF5 Setup

The EF5 was set up over the CONUS to run CREST, SAC-SMA and HP water balance models all coupled with KW routing. No snow module was used for these simulations. The modeling domain was set to exactly match the MRMS domain with a regular 0.01° grid spanning from –130.0 to –60.0 longitude and 20.0 to 55.0 latitude. This grid was picked to fully exploit the resolution provided by the MRMS precipitation estimates. The basic files, which are the DEM, FDR, and FAM, were derived from the NED (Gesch et al., 2009). The NED data was resampled to the 0.01° resolution using an arithmetic mean and then FDR and FAM were derived using ESRI ArcGIS and the ArcHydro toolbox. A priori distributed parameter maps are preferred where available and as such were used for impervious area and soil parameters in the hydrologic models. The models were run uncalibrated because there is a focus on providing information over the CONUS to improve flash flood warnings in overland areas which are not typically instrumented with gauges or adequately modeled through traditional regionalization approaches.

The CREST parameters used for this study are largely based on a-priori maps of soil information generated by Miller and White (1998) utilizing the STATSGO dataset. Table 1 summarizes the EF5/CREST parameters and the values used in this study. The $b$ parameter was derived from the soil texture map provided by Miller and White (1998) with a lookup table from Cosby et al. (1984) then used to convert from the soil texture into the exponent parameter. The lookup table for $b$ is provided in Table 2. The $W_m$ parameter map was generated from resampling the available water capacity 250 cm depth map in Miller and White (1998) to the domain used here with bilinear interpolation. The $F_c$ parameter for EF5/CREST was produced using the permeability map from Miller and White (1998). The percent impervious area was derived from the NLCD 2011 edition impervious area from Xian et al. (2011) resampled using average interpolation onto the study domain. Figure 4 shows the spatial distributions of the non-uniform parameter values over the CONUS. The $K_e$ and $I_{wu}$ are the only EF5/CREST parameters without distributed a-prior parameters.

The EF5/SAC-SMA parameters were taken directly from work done by Zhang et al. (2011) because this work is most comparable to what is used operationally by the NWS. Table 3 lists the parameters and their respective values used in this





**Table 1.** Description of parameters used in the CREST water balance model

| Parameter | Description | Value | Grid Source | Min | Mean | Max |
|---|---|---|---|---|---|---|
| WM | Water capacity of soil in mm | 1 | Miller and White (1998) | 0 | 206 | 2500 |
| FC | Saturated hydrologic conductivity in mm h$^{-1}$ | 1 | Miller and White (1998) | 0 | 8 | 51 |
| B | Exponent of the infiltration curve | 1 | Miller and White (1998) | 0 | 5 | 12 |
| IM | Percentage impervious area in % | 1 | Xian et al. (2011) | 0 | 1 | 96 |
| KE | Potential evapotranspiration adjustment factor | 1 | NA | NA | NA | NA |
| IWU | Initial soil water content | 75 | NA | NA | NA | NA |

**Table 2.** Soil Texture and EF5/CREST $b$ parameter value as derived from Cosby et al. (1984)

| Soil Texture | $b$ |
|---|---|
| Sandy loam | 4.74 |
| Sand | 2.79 |
| Loamy sand | 4.26 |
| Loam | 5.25 |
| Silty loam | 5.33 |
| Sandy clay loam | 6.77 |
| Clay loam | 8.17 |
| Silty clay loam | 8.72 |
| Sandy clay | 10.73 |
| Silty clay | 10.39 |
| Light clay | 11.55 |

study. The PCTIM, ADIMP, SIDE, and RIVA parameters are using lumped values defined in the tables because a-priori grids are not available.

The kinematic wave parameters used by EF5 are listed in Table 4. These parameter values are used for all model combinations when coupled with CREST, SAC-SMA and HP water balance options for this study. The parameters are a priori-based on statistical relationships with geomorphological, precipitation and soil parameters developed in Vergara et al. (2016). Observed $\alpha$ and $\beta$ values were computed from the cross-sections and discharge values measured by the USGS. These observed values were then modeled using GAMLSS which allows for the extrapolation of information collected at the approximately 10,000 USGS discharge stations in the CONUS to everywhere on the hydrologic model grid. The parameters used are basin area, elongation ratio, relief ratio, slope index, slope to outlet, mean annual precipitation, mean annual temperature, K factor, depth-to-rock, rock volume percentage, soil texture, curve number, and river length.





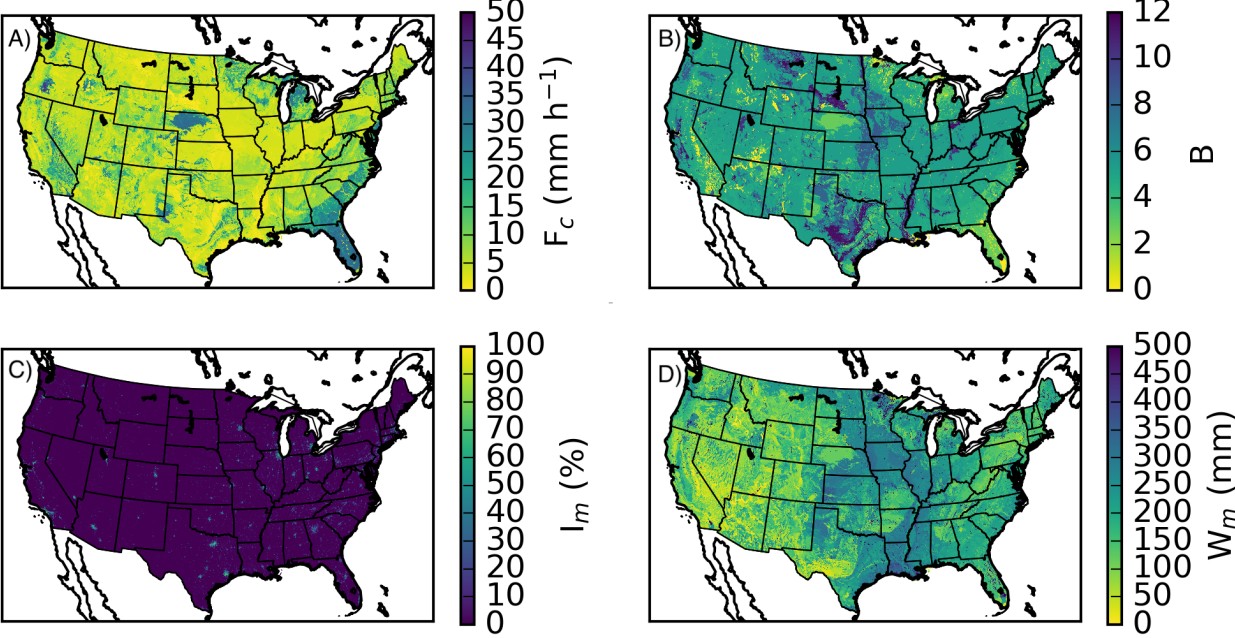

**Figure 4.** The EF5/CREST A) $F_c$, B) $b$, C) $I_m$, and D) $W_m$ distributed parameters used in the study over the conterminous U.S..

The $\alpha_0$ parameter was computed using Manning's equation for overland flow:

$$\alpha_0 = \frac{1}{n} S^{\frac{1}{2}} \tag{25}$$

where the $S$ is the slope computed from DEM, $n$ is Manning's roughness coefficient. The roughness coefficient was computed from UMD MODIS land cover type mosaics (Channan et al., 2014) and a lookup table from Chow et al. (1988) documented in Table 5. Figure 5 shows the resulting kinematic wave parameter maps for the CONUS. The parameters have clear signs of influence from the geomorphological information used to derive them.

EF5 was run for the period from 2001 through 2011 for USGS stream gauges with a basin area under 1,000 km². There are 4,366 stream gauges over the CONUS that meet this basin area threshold. The MRMS reanalysis precipitation rates with a time step of five minutes were used as the precipitation forcing for EF5. The PET data was climatological monthly mean data derived in Koren et al. (1998). EF5 was run with a five minute time step producing five minute output simulated time series. The resulting simulations took 1-week of computer time for the EF5/CREST combination and 2.5-weeks of computer time for EF5/SAC-SMA illustrating the relative differences in complexity and performance between the two water balance models. The year 2001 was used as a model warmup period and so results will only be presented from 2002 through 2011.





**Table 3.** Description of parameters used in the SAC-SMA water balance model

| Parameter | Description | Value | Grid Source |
|---|---|---|---|
| UZTWM | Upper zone tension water capacity in mm | 1 | Zhang et al. (2011) |
| UZFWM | Upper zone free water capacity in mm | 1 | Zhang et al. (2011) |
| UZK | Depletion rate from upper zone free water storage from interflow in day$^{-1}$ | 1 | Zhang et al. (2011) |
| LZTWM | Lower zone tension water capacity in mm | 1 | Zhang et al. (2011) |
| LZFSM | Lower zone supplemental free water capacity in mm | 1 | Zhang et al. (2011) |
| LZFPM | Lower zone primary free water capacity in mm | 1 | Zhang et al. (2011) |
| LZSK | Rate of depletion of the lower zone supplemental free water storage in day$^{-1}$ | 1 | Zhang et al. (2011) |
| LZPK | Rate of depletion of the lower zone primary free water storage in day$^{-1}$ | 1 | Zhang et al. (2011) |
| ZPERC | Maximum and minimum percolation rate ratio | 1 | Zhang et al. (2011) |
| REXP | Shape parameter of the percolation curve | 1 | Zhang et al. (2011) |
| PFREE | Percolation fraction that goes directly to the lower zone free water storage | 1 | Zhang et al. (2011) |
| PCTIM | Percentage impervious area in % | 0.1 | NA |
| ADIMP | Maximum fraction of additional impervious area from saturation | 0.1 | NA |
| RIVA | Riparian vegetation fractional area | 1.0 | NA |
| SIDE | Ratio of deep percolation from lower zone free water storage | 0.0 | NA |
| ADIMC | Initial additional impervious area from saturation | 1.0 | NA |
| UZTWC | Initial filled amount of upper zone tension water | 0.55 | NA |
| UZFWC | Initial filled amount of upper zone free water | 0.14 | NA |
| LZTWC | Initial filled amount of lower zone tension water | 0.56 | NA |
| LZFSC | Initial filled amount of lower zone supplemental free water | 0.11 | NA |
| LZFPC | Initial filled amount of lower zone primary free water | 0.46 | NA |

**Table 4.** Description of parameters used in kinematic wave routing scheme.

| Parameter | Description | Value | Grid Source | Min | Mean | Max |
|---|---|---|---|---|---|---|
| $\alpha$ | Kinematic wave multiplier coefficient | 1 | Vergara et al. (2016) | 0.4 | 3 | 149 |
| $\beta$ | Kinematic wave power coefficient | 1 | Vergara et al. (2016) | 0.4 | 0.7 | 1.0 |
| $\alpha_0$ | Kinematic wave conveyance parameter for overland | 1 | Vergara et al. (2016) | 0.06 | 0.7 | 18 |
| Under | Subsurface flow speed in m s$^{-1}$ | 0.0001 | Miller and White (1998) | 0 | 8 | 51 |
| LeakI | Reduction in interflow storage in % | 1 | Zhang et al. (2011) | 0.127 | 0.128 | 0.129 |
| Th | Drainage area threshold for channel cells | 10 | NA | NA | NA | NA |
| ISU | Initial water storage in channel grid cells | 0.0 | NA | NA | NA | NA |





**Table 5.** Description of overland flow parameterizations.

| UMD Class | Description | Manning's $n$ |
|---|:---:|---|
| 0 | Water | 0.001 |
| 1 | Evergreen Needleleaf Forest | 0.1 |
| 2 | Evergreen Broadleaf Forest | 0.1 |
| 3 | Deciduous Needleleaf Forest | 0.1 |
| 4 | Deciduous Broadleaf Forest | 0.1 |
| 5 | Mixed Forest | 0.1 |
| 6 | Woodland | 0.1 |
| 7 | Wooded Grassland | 0.3 |
| 8 | Closed Shrubland | 0.3 |
| 9 | Open Shrubland | 0.2 |
| 10 | Grassland | 0.17 |
| 11 | Cropland | 0.035 |
| 12 | Bare Ground | 0.01 |
| 13 | Urband and Built | 0.015 |





**Figure 5.** Spatial distributions of the kinematic wave parameters used over the CONUS in this study.





## 3.2 CONUS Bulk Simulation Validation

A bulk analysis was performed to evaluate the skill of the modeling system at every USGS gauge with a basin area less than 1,000 km$^2$. The time series from the EF5 simulations can be evaluated as a function of the performance at each individual stream gauge. This information can then be viewed in bulk to gather of a sense of how the system performs spatially in terms

of the overall mass of water, and the correlation between simulated and observed events. The accuracy of the simulations is judged using Pearson's linear correlation coefficient, $CC$, defined as:

$$CC = \frac{\mathbf{Cov}(Q_{sim}, Q_{obs})}{\sqrt{\mathbf{Var}(Q_{sim})\mathbf{Var}(Q_{obs})}} \qquad (26)$$

where $Q_{sim}$ is the simulated discharge value and $Q_{obs}$ is the USGS measured discharge value. The values for correlation coefficient can range from –1 to 1 with 1 being the best. The normalized bias of the simulations is computed using:

$$bias = \frac{\sum_{i=1}^{N}(Q_{sim}^i - Q_{obs}^i)}{\sum_{i=1}^{N} Q_{obs}^i} * 100 \qquad (27)$$

where $N$ is the number of observations in the discharge time series. Normalized bias ranges from –100 % to $\infty$ with 0 % being the best. Finally the NSE (Nash and Sutcliffe, 1970), commonly used as a skill metric to define simulations that have better skill than the mean of the observations would have, is computed as:

$$nse = 1 - \frac{\sum_{i=1}^{N}(Q_{sim}^i - Q_{obs}^i)^2}{\sum_{i=1}^{N}(Q_{obs}^i - \overline{Q_{obs}})^2} \qquad (28)$$

where $\overline{Q_{obs}}$ is the mean of the discharge observations for this station. The values for NSE range from $-\infty$ to 1 with 1 being a simulation perfectly matching the observations.

Figure 6 shows the spatial distribution of NSE, CC, and normalized bias for the three simulations. The maximum, median, and minimum values for NSE, CC, and bias are summarized in Table 6. Overall, the water balance modules yield comparable performance with a few notable patterns. There is a notable drop in accuracy and negative bias in the Intermountain West region

according to all three models. The relatively poor performance here is due to inaccurate precipitation forcings. First, radar-based precipitation estimates face challenges due to intervening blockages by the mountains and greater distances between radars (Maddox et al., 2002). Second, there is a large portion of precipitation that falls as snow in this high-elevation region. While the parent MRMS precipitation forcings separate frozen and liquid precipitation, EF5 did not consider snow processes in this study. As such, results in these regions should be used with caution when frozen precipitation processes are active.







**Figure 6.** The spatial distribution of NSE, CC, and normalized bias computed against USGS observations for all gauges with drainage areas less than 1,000 km$^2$ as a function of the water balance module.

The results from this study using EF5/CREST, EF5/SAC-SMA, and EF5/HP all coupled with kinematic wave routing and a-priori uncalibrated parameters for all models are acceptable showing no significant systematic errors as a function of watershed scale. EF5 is able to ingest MRMS five minute precipitation rate files and a ten-year simulation completed in a week of computer time is a reasonable expectation given the high resolution of the basic grids. The overall skill of the system is reasonable given its uncalibrated nature and on some watersheds the skill is equivalent to that expected for a calibrated hydrologic model. The results in Figure 7 show no significant trend in accuracy versus basin area for the range of flash flood basins from 1 km$^2$ to 1,000 km$^2$. The EF5/HP model works as a "worst case scenario" and exhibits large positive bias for most watersheds which is expected behavior for a completely impervious land surface. The EF5/HP model provides an upper envelope when used as

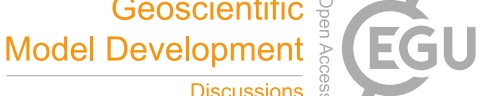

**Table 6.** Statistical Summary of EF5 Performance

|  | EF5/CREST | EF5/SAC-SMA | EF5/HP |
|---|---|---|---|
| Max NSE | 0.71 | 0.76 | 0.59 |
| Median NSE | –0.06 | –0.03 | 0.08 |
| Min NSE | –313 | –613 | –20 |
| # basins NSE > 0 | 1,825 | 1,982 | 3,642 |
| Max CC | 1.0 | 0.92 | 0.83 |
| Median CC | 0.40 | 0.35 | 0.36 |
| Min CC | –0.47 | –1.0 | –0.25 |
| Median Bias | 9 % | –8 % | 248 % |

a member of an ensemble which is useful for diagnosing errors in precipitation input forcing, approximating the behavior of runoff on burn areas, and diagnosing situations in which the soils are completely saturated.

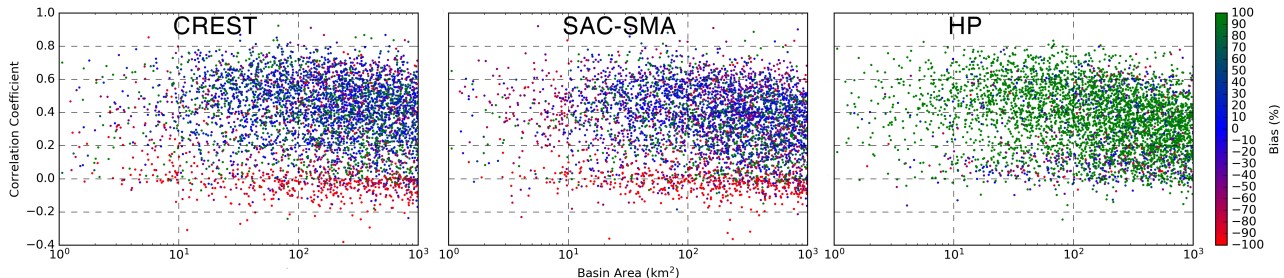

**Figure 7.** The CC and normalized bias versus contributing drainage area for all gauges with drainage areas less than 1,000 km$^2$ as a function of the water balance module.

## 4 Summary & Future Work

To further the goal of producing accurate, precise, and timely flash flood warnings while utilizing new precipitation datasets, a
5 new high-resolution distributed hydrologic modeling platform, EF5, was created to facilitate this process. EF5 features flexible options for choosing which water balance models and routing schemes to simulate with or run all of them to generate a hydrologic ensemble. The resulting software package was used for generating 5 minute simulations for 4,366 gauge locations across the CONUS with uncalibrated, a-priori parameters for the EF5/CREST, EF5/SAC-SMA, and EF5/HP water balance models coupled to kinematic wave routing. Furthermore, EF5 is being used for training, capacity building and operational
10 forecasting (Clark et al., 2017). EF5/CREST and EF5/SAC-SMA run with uncalibrated, a-priori parameters over the CONUS and MRMS precipitation forcing produce skillful simulations except for in mountainous regions with NSE scores up to 0.76.



EF5/HP produces useful estimates for "worst case scenarios" if all rainfall is converted into runoff such as over burn areas, heavily urbanized watersheds, or situations in which the soils are saturated. Differences between EF5/HP and those of the EF5/CREST & EF5/SAC-SMA model runs illustrate the inherent uncertainties with hydrologic model parameter estimation and the non-linear conversion from rainfall into runoff.

The future for EF5, hydrologic modeling and developing climatologies of flash floods is extremely promising. EF5 is being used to power the distributed hydrologic models in the FLASH system (Gourley et al., 2017) where NWS forecasters are using it in a warning decision support role. The operational version of EF5 runs across the conterminous U.S. and territories at 1-km spatial resolution and frequency of every 10 min. Future developments for EF5 may include diffusive wave routing to better handle shallow slope basins, and a parameterization for reservoirs so that they can also be accommodated. EF5 currently has a snow module, but a-priori parameter development is required before it can be deployed across the CONUS and globally. Continued improvements to EF5 are a must to ensure it remains accessible to all users in the future. A better graphical user interface on the Windows operating system may improve classroom and workshop usability. Solutions for containerizing EF5 such as Docker should be explored to see if there are significant advantages to this workflow.

In the future, new observational platforms will be necessary to collect the observations needed to validate distributed hydrologic models. As the spatiotemporal resolution the hydrologic models are simulating decreases the need for observations to help validate the models increases. These new observations could come from augmentations of existing datasets such as with stream radars that can map the channel cross-section, water velocity and water height. Unpiloted aerial systems have a promising role in the future as well, an automated platform that maps out flood waters in real time would be invaluable as a dataset for verifying hydrologic models.

*Code availability.* The source code to EF5 is available on GitHub at https://github.com/HyDROSLab/EF5 and documented in Flamig et al. (2016). EF5 is released into the public domain for all use cases. Documentation, including the user manual and training videos, can be found at http://ef5.ou.edu.

*Author contribution.* First author Flamig developed the water balance schema within the Ensemble for Flash Flood Forecasting Framework and conducted the model reanalyses and evaluations shown herein. Second author Vergara assisted in the development of the models' a-prior parameters and developed the parameterizations for the kinematic wave routing scheme. Third author Gourley managed the project and assisted in the writing of the manuscript.

*Competing interests.* The authors declare no competing interests.

*Acknowledgements.* Funding for this research was provided by the Disaster Relief Appropriations Act of 2013 (P.L. 113-2), which provided support to the Cooperative Institute for Mesoscale Meteorological Studies at the University of Oklahoma under Grant NA14OAR4830100.



The authors would like to thank Race Clark who contributed significantly to the development of training materials for EF5 and provided valuable feedback that materially improved the software. The authors also thank numerous undergraduate students who provided feedback and bug reports on EF5 while using it in an educational setting. We also thank Faith Mitheu, the staff working on SERVIR at the Regional Centre For Mapping Resource For Development, and the staff at Hydrological Services Namibia for valuable feedback which led to an
5    improved hydrologic modeling system.





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
