# Peer review of "The Ensemble Framework For Flash Flood Forecasting (EF5) v1.2: Description and Case Study"

_Geoscientific Model Development, 2020_

## Referee Comment (RC1) · Seann Reed (Referee) · 1 Jun 2020

GENERAL COMMENTS:

This paper provides a pragmatic look at three models which can provide nation-wide flash flood model guidance in the EF5 framework. While there are certainly theoretical limitations with the modeling approaches, it is commendable that the authors and developers have forged ahead with this approach to make it available to operational forecasters. The description of the modeling framework, parameter estimation, and analysis is an important contribution to the literature. The bulk, high-level results show that additional work is needed to develop recommendations to forecasters on whether

to rely more on the CREST or SAC models. Additional work to determine where to invest in model enhancements would also be beneficial.

SPECIFIC COMMENTS:

Somewhere there should be mention of the National Water Model and how the EF5 differs (e.g. temporal scale) and thus provides forecast information not available from the NWM.

I have some concern about applying the SAC-SMA model at increasingly smaller grid scales, particularly if the same a-priori parameters are used. I've seen 'good' results from 4 km2 and 16 km2 gridded SMA applications but there was also considerable improvement from calibration at these scales. There is definitely a scale dependency in the SAC-SMA model (. (Finnerty, B.D., Smith, M.B., Seo, D.-J., Koren, V., Moglen, G.E., 1997. Space-time sensitivity of the Sacramento model to radar-gauge precipitation inputs. Journal of Hydrology, Vol. 203, 21-38.). Also, the gridded SAC-SMA implementation assumes baseflow is an independent process by grid cell (no cell-to-cell soil water exchange). This assumption becomes less plausible at smaller grid cells. However, I still think that your application at about a 1 km2 scale is still worthy of evaluation in this context.

At the top of p.13 the authors state that the subsurface discharge is routed through linear reservoirs rather than using kinematic wave. That sounds like a reasonable assumption, but I did not see an explanation in the paper as to how the linear reservoir parameters are estimated.

P.14 – Why not derive a grid of PCTIM from the NLCD like you did with the comparable CREST parameter?

p. 19 – I would recommend using Snow17 if this analysis will be redone at any point in the future. The authors note 'As such, results in these regions should be used with caution when frozen precipitation processes are active.' I would not be surprised if ex-

cluding the Snow model also has some impacts on the relative performanc of SAC and CREST in the Northwest, North Central and North Eastern US. Without snow retention, simulated spring runoff could be sharper than what really occurs or there could be winter-time simulated events that don't really occur. Also, not modeling the effects of frozen ground in the North Central US could result in springtime under-simulations of events. Due to structural differences, the CREST and SAC-SMA would likely react relatively differently to changed rain-plus-melt series compared to how they react in your current study. I believe the last version of the HL-RDHM I saw was delivered with a-priori estimates of the major Snow17 parameters and provides guidance on estimating the additional parameters needed for a basic simulation.

TECHNICAL CORRECTIONS:

p2, line 10: 'high resolution forward hydrologic simulation'– take out the word 'forward'

p3, line 26: Add comma 'Given the evidence above,'

p3, line 32: Add comma 'resolution, necessitating'

p4, line 27: 'TRMM, and TMPA. . .'

p4, line 27: "While . . ." This is not a sentence. Could delete "While" or just delete the sentence altogether.

p.5, Delete the sentence "For completeness the base classes for the routing and snow components are included below."

p.5, third from last line, should say '. . .takes fast and slow components. . .'

p6, line 22: I suggest 'followed' rather than 'proceeded'

p6, line 24: is a 'derivative of'

p6, line 26: Wang et al. (2011) documented the first version of CREST. . .

p8, line 12: should be "coarser" instead of "courser"

p10, line 2: is 'classified' as

p 12, line 3: No need for () around Ponce, 1991

p.22, line 15: Should say "...as model resolutions are increasing the need for validating observations also increases."

―――――――――――――――――

---

## Referee Comment (RC2) · Anonymous Referee #2 · 8 Jul 2020

This manuscript describes the EF5 that allows to produce hydrological runoff outputs (e.g., discharge) by i) adapting different inputs for precipitation (e.g. from multi-radar multi-sensor MRMS for the presented case over the CONUS) and ii) combing (as an "ensemble" of) existing algorithms of snow melt (not presented details here), water balance, routing, and calibration (not used in the presented case).

The method (Section 2) focuses on details mostly three water balance models and routing parts of EF5 and case analyses for the evaluation (Section 3), which were parts of the author's Ph.D. dissertation published in 2016 with Open Access: https://shareok.org/handle/11244/44865, e.g., Chapter 3 and some parts in Chapter 2

with major duplications of figures, texts and the presented cases.

The submission of theses (unpublished yet to another peer-review journal) is in general encouraged. However, I found the method and evaluation of EF5 presented here is not sufficient to fulfill the key scope of GMD (e.g. reproductivity of the work). Here, this reproductivity is very briefly mentioned in summary and future section; e.g., implementations for flash flood forecasting within the FLAHS project (cited briefly in P22, L3-4; Gourley et al. 2017) and at Namibia (P21, L9-10; Clark et al. 2017). However, it should be better addressed by adding discussions and implemented case summaries in this manuscript as well. So, I do not recommend its publication without a major revision considering following points that may help the manuscript to be more interesting and updated. (Note: P- page, L- line number in each page)

1. The code uploaded in the provided link (https://github.com/HyDROSLab/EF5, Flamig, Z. L., Vergara, H., Clark, III, R., Hong, Y., and Gourley, J. J.: EF5: Version 1.0, doi:10.5281/zenodo.59123, http://dx.doi.org/10.5281/zenodo.59123, 2016) is indeed v1.0 not v1.2 that is indicated in the title. If there is any update in the code and manual, please comment them in the text. Also, I found the following version by the same author but under the name of "training", would this example can be presented in this paper as well? Zac Flamig. (2018, March 13). HyDROSLab/EF5: More bug fixes. (Version v1.2.3). Zenodo. http://doi.org/10.5281/zenodo.1197006 The manual exists in Latex file but pdf can be also appreciated.

2. Although the name of the model contains "for flash flood forecasting" and the abstract says "the results of the study show that the three uncalibrated water balance models linked to kinematic wave routing are skillful in streamflow prediction", the presented method and analyses hardly contain any predicted outputs ahead in time. The evaluation is also done only in terms of the discharge assessment (every 5 minutes at USGS gauge points in near real time precipitation forcing). Abstract should reflect better what has been presented in this work. Adding more examples from the implementational works including detail limitations will also make the manuscript more solid;

e.g., P6 L2-3 and P13 L17-19 given that EF5 is now operational over CONUS.

3. It is not clear that how important adding "Snow (melt) component" in EF5; this seems a newly added feature to EF5 (introduction e.g., P4, L5-7), yet the detail background/examples were not presented in the method. Also, the interpretation of the presented cases (P19, L22-24 linked to the not-used "snow module") needs more solid evidences. What kind of caution (or a priori parameter development as mentioned in P22, L10) should be considered by the users? Please explain more explicitly.

Minor comments

1. The reference link was broken - Flamig, Z. L., Vergara, H., Clark, III, R., Hong, Y., and Gourley, J. J.: EF5: Version 1.0, doi:10.5281/zenodo.59123, http://dx.doi.org/10.5281/zenodo.59123, 2016.

2. Some acronyms need to be better informed: e.g.,P13, KW, NED, P14 GAMLSS

3. Table1, fix parameters the same as written in P9, IWU has no unit? Check units in other tables as well.

4. P6, 20-22: add reference or provide evidence.

5. P20, L1-3, L4-5, L8, P21 L1-2: Need better explanations.

6. P22, L2-4: Provide more clear explanation and supporting materials in the results.

7. P22, L15-16, It is not clearly written. Revise the sentence.

---

## Author Comment (AC1) · 10 Aug 2020

article [utf8]inputenc [dvipsnames]xcolor

[Figure]

**Response to Interactive comment on "The Ensemble Framework For Flash Flood Forecasting (EF5) v1.2: Description and Case Study" by Zachary L. Flamig et al. from Seann Reed (Referee)**

August 10, 2020

**General comments**

**Reviewer.** This paper provides a pragmatic look at three models which can provide nation-wide flash flood model guidance in the EF5 framework. While there are certainly theoretical limitations with the modeling approaches, it is commendable that the authors and developers have forged ahead with this approach to make it available to operational forecasters. The description of the modeling framework, parameter estimation, and analysis is an important contribution to the literature. The bulk, high-level results show that additional work is needed to develop recommendations to forecasters on whether to rely more on the CREST or SAC models. Additional work to determine where to invest in model enhancements would also be beneficial.

**Response.** We appreciate the constructive comments supplied by this expert re-
viewer. Below, we address each comment and provide revisions that were made to the manuscript. We believe an improved manuscript has resulted and we are thankful.

**Reviewer. Specific comments**

**Reviewer.** Somewhere there should be mention of the National Water Model and how the EF5 differs (e.g. temporal scale) and thus provides forecast information not available from the NWM.

**Response.** Agreed. The National Water Model is operational within NOAA and provides new hydrologic forecasting capabilities at ungauged grid points, similar to EF5. There are differences in the model structures, forcings, etc., but the largest differentiation between them is the frequency and latency at which forecast products are available. The NWM runs at the top of the hour and it takes approximately 30-45 min. for a simulation across the conterminous US to complete. In the case of EF5, forecasts are launched every 10 min., and are completed in less than 10 min. This means the maximum latency with NWM products is 1 hr 44 min and approximately 19 min with EF5. For these reasons, EF5 is more applicable to monitoring and forecasting of conditions at the flash flood scale (including pluvial, overland flooding), while NWM forecasts can apply to larger streams and rivers. Nevertheless, EF5 and NWM both serve to advance the tools available to NWS forecasters by supplying hydrologic model forecasts at ungauged grid cells across the U.S. and territories. We have now added the following text on Pg. 3. Line 28 in regard to this comment:

More recently, the U.S. NWS implemented the National Water Model (NWM), which is a variant of the Weather Research and Forecasting Model Hydrological modeling system (WRF-Hydro) (Gochis et al., 2014). This modeling framework is more holistic in that it is being developed to address multiple hydrologic applications ranging from water resources management, stream temperature forecasting, coupling to storm surge models for coastal flooding applications, surface and groundwater interactions, and channel

losses in semi-arid environments. The wide range of applications requires more model complexity and thus the framework utilizes the Noah-Multiparameterization Land Surface Model (NOAH-MP) as its core. The utility of the NWM for flash flood forecasting will require sub-hourly data latency, yet there has been some recent progress on applications (Viterbo et al., 2020).

**Reviewer.** I have some concern about applying the SAC-SMA model at increasingly smaller grid scales, particularly if the same a-priori parameters are used. I've seen 'good' results from 4 km2 and 16 km2 gridded SMA applications but there was also considerable improvement from calibration at these scales. There is definitely a scale dependency in the SAC-SMA model (. (Finnerty, B.D., Smith, M.B., Seo, D.-J., Koren, V., Moglen, G.E., 1997. Space-time sensitivity of the Sacramento model to radar-gauge precipitation inputs. Journal of Hydrology, Vol. 203, 21-38.). Also, the gridded SAC-SMA implementation assumes baseflow is an independent process by grid cell (no cell-tocell soil water exchange). This assumption becomes less plausible at smaller grid cells. However, I still think that your application at about a 1 km2 scale is still worthy of evaluation in this context.

**Response.** These are valid points. We should note that SAC-SMA model structure was adopted within EF5 largely given its legacy within the National Weather Service. There are many users of the model in the NWS who are experienced with it. EF5 was implemented at a grid cell resolution of 1 km$^2$ in line with the resolution of the MRMS radar-based rainfall forcing. Thus, a choice was made to implement the model as closely as possible to its original structure without the requirement of re-deriving all the parameters.

**Reviewer.** At the top of p.13 the authors state that the subsurface discharge is routed through linear reservoirs rather than using kinematic wave. That sounds like a reasonable assumption, but I did not see an explanation in the paper as to how the linear reservoir parameters are estimated.

**Response.** Thanks for pointing this out. We have added the following text on 15, Line 6:

"Estimates for linear reservoir model parameters Under and LeakI for subsurface flow are based on *Fc* (hydraulic conductivity) and SAC-SMA's UZK parameter (Table 3) respectively, using conversion factors for units consistency."

**Reviewer.** P.14 – Why not derive a grid of PCTIM from the NLCD like you did with the comparable CREST parameter?

**Response.** We chose to concentrate new model developments in the CREST model given our experience with it, and decided to implement SAC-SMA as closely as possible to its original construction. Otherwise, the model would have deviated too much from its original implementation, from which NWS forecasters have familiarity and experience.

**Reviewer.** p. 19 – I would recommend using Snow17 if this analysis will be redone at any point in the future. The authors note 'As such, results in these regions should be used with caution when frozen precipitation processes are active.' I would not be surprised if excluding the Snow model also has some impacts on the relative performanc of SAC and CREST in the Northwest, North Central and North Eastern US. Without snow retention, simulated spring runoff could be sharper than what really occurs or there could be winter-time simulated events that don't really occur. Also, not modeling the effects of frozen ground in the North Central US could result in springtime under-simulations of events. Due to structural differences, the CREST and SAC-SMA would likely react relatively differently to changed rain-plus-melt series compared to how they react in your current study. I believe the last version of the HL-RDHM I saw was delivered with apriori estimates of the major Snow17 parameters and provides guidance on estimating the additional parameters needed for a basic simulation.

**Response.** We agree. The initial operational capability of EF5 did not include the

Snow17 module even though it is a component supported within the framework. At the time we implemented EF5 into NWS operations, we had little experience with Snow17 and didn't feel it had the level of validation in flash flood forecasting as the other core modeling components. As such, it didn't reach the technical readiness levels sufficient for transitioning to operations at that time. That being said, we have noted shortcomings in spring runoff responses in the northern tier of the U.S, just as you suggested. This was particularly true following the exceptionally high and late snowpack during the spring of 2019. Some forecasters noted more muted responses from EF5 given their observations of flooding. In short, soil saturation was underestimated in the models and this resulted in underforecasts of surface runoff. We are presently addressing this issue by including Snow17 in simulations and also assimilating observations and model states from NLDAS and NWM into EF5 to improve soil moisture simulations. Early results indicate improvements in doing this. We intend to transition these modules to operations with version 20 (currently we're transitioning MRMS v12). But, we felt this information was not sufficiently validated to include in the manuscript. To improve clarity of the study's intentions, we added the following statement on Pg. 13, Line 15: "The intention of this study is to evaluate the accuracy of the model version that was transitioned to the NWS as part of the EF5 initial operational capability."

**Technical corrections**

**Reviewer.** p2, line 10: 'high resolution forward hydrologic simulation'– take out the word 'forward'

**Response.** Done.

**Reviewer.** p3, line 26: Add comma 'Given the evidence above,'

**Response.** Done.

**Reviewer.** p3, line 32: Add comma 'resolution, necessitating'

**Response.** Done

**Reviewer.** p4, line 27: 'TRMM, and TMPA. . .'

**Response.** Changed to just TMPA as we accidentally expanded TRMM from the TMPA acronym.

**Reviewer.** p4, line 27: "While . . ." This is not a sentence. Could delete "While" or just delete the sentence altogether.

**Response.** Deleted.

**Reviewer.** p.5, Delete the sentence "For completeness the base classes for the routing and snow components are included below."

**Response.** Done.

**Reviewer.** p.5, third from last line, should say '. . .takes fast and slow components. . .'

**Response.** Changed.

**Reviewer.** p6, line 22: I suggest 'followed' rather than 'proceeded'

**Response.** Changed.

**Reviewer.** p6, line 24: is a 'derivative of'

**Response.** Changed.

**Reviewer.** p6, line 26: Wang et al. (2011) documented the first version of CREST. . .

**Response.** Done.

**Reviewer.** p8, line 12: should be "coarser" instead of "courser"

**Response.** Done.

**Reviewer.** p10, line 2: is 'classified' as

**Response.** Done.

**Reviewer.** p 12, line 3: No need for () around Ponce, 1991

**Response.** Corrected.

**Reviewer.** p.22, line 15: Should say "...as model resolutions are increasing the need for validating observations also increases."

**Response.** Sentence has been changed to the following: "As the spatiotemporal resolution of hydrologic models is increasing, the need for validating observations also increases."

---

## Author Comment (AC2) · 10 Aug 2020

article [utf8]inputenc [dvipsnames]xcolor

**Response to Interactive comment on "The Ensemble Framework For Flash Flood Forecasting (EF5) v1.2: Description and Case Study" by Zachary L. Flamig et al. from Anonymous Referee #2**

August 10, 2020

**General comments**

**Reviewer.** This manuscript describes the EF5 that allows to produce hydrological runoff outputs (e.g., discharge) by i) adapting different inputs for precipitation (e.g. from multi-radar multi-sensor MRMS for the presented case over the CONUS) and ii) combing (as an "ensemble" of) existing algorithms of snow melt (not presented details here), water balance, routing, and calibration (not used in the presented case). The method (Section 2) focuses on details mostly three water balance models and routing parts of EF5 and case analyses for the evaluation (Section 3), which were parts of the author's Ph.D. dissertation published in 2016 with Open Access: https://shareok.org/handle/11244/44865, e.g., Chapter 3 and some parts in Chapter 2 with major duplications of figures, texts and the presented cases.

**Response.** We appreciate the comments supplied by this reviewer. Indeed, a number

of figures and results are adapted from the first author's dissertation work completed at the University of Oklahoma. Below, we respond to each comment and cite instances in which there are revisions made to the manuscript.

**Reviewer.** The submission of theses (unpublished yet to another peer-review journal) is in general encouraged. However, I found the method and evaluation of EF5 presented here is not sufficient to fulfill the key scope of GMD (e.g. reproductivity of the work). Here, this reproductivity is very briefly mentioned in summary and future section; e.g., implementations for flash flood forecasting within the FLAHS project (cited briefly in P22, L3-4; Gourley et al. 2017) and at Namibia (P21, L9-10; Clark et al. 2017). However, it should be better addressed by adding discussions and implemented case summaries in this manuscript as well. So, I do not recommend its publication without a major revision considering following points that may help the manuscript to be more interesting and updated. (Note: P- page, L- line number in each page)

**Response.** We have added the input configurations and variables to GitHub at https://github.com/HyDROSLab/EF5-US-Parameters to improve the reproducibility of the test case studies conducted here. The GMD guidelines stipulate that, "The scientific goal is reproducibility: ideally, the description should be sufficiently detailed to in principle allow for the re-implementation of the model by others, so all technical details which could substantially affect the numerical output should be described." The standard as we understand it is that someone should be able to take this paper and create their own version of the model which can be run to produce scientifically similar results (i.e. explicitly not bitwise reproducibility). As noted in the GMD guidelines, we are supporting the model description with "summary outputs from test case simulations" that are not meant to be exhaustive of all configurations, modes of operation, and modules included in EF5. Nevertheless, the availability of model forcings, parameters, code, documentation, and training materials satisfy the reproducibility requirement for the journal.

On Pg. 22, Line 21, we have added the following statement in response to this comment: "The spatially distributed DEM, routing, and surface water balance parameters as well as potential evapotranspiration forcings are available at https://github.com/HyDROSLab/EF5-US-Parameters."

**Reviewer.** 1. The code uploaded in the provided link (https://github.com/HyDROSLab/EF5, Flamig, Z. L., Vergara, H., Clark, III, R., Hong, Y., and Gourley, J. J.: EF5: Version 1.0, doi:10.5281/zenodo.59123, http://dx.doi.org/10.5281/zenodo.59123, 2016) is indeed v1.0 not v1.2 that is indicated in the title. If there is any update in the code and manual, please comment them in the text. Also, I found the following version by the same author but under the name of "training", would this example can be presented in this paper as well? Zac Flamig. (2018, March 13). HyDROSLab/EF5: More bug fixes. (Version v1.2.3). Zenodo. http://doi.org/10.5281/zenodo.1197006 The manual exists in Latex file but pdf can be also appreciated.

**Response.** We have updated the DOI (http://doi.org/10.5281/zenodo.569078) to correctly point to version 1.2 in the text. EF5 is under active development so there are newer versions than presented here in this paper.

**Reviewer.** 2. Although the name of the model contains "for flash flood forecasting" and the abstract says "the results of the study show that the three uncalibrated water balance models linked to kinematic wave routing are skillful in streamflow prediction", the presented method and analyses hardly contain any predicted outputs ahead in time. The evaluation is also done only in terms of the discharge assessment (every 5 minutes at USGS gauge points in near real time precipitation forcing). Abstract should reflect better what has been presented in this work. Adding more examples from the implementational works including detail limitations will also make the manuscript more solid;

e.g., P6 L2-3 and P13 L17-19 given that EF5 is now operational over CONUS.

**Response.** The name of the tool reflects the goal and overall utility of the framework in association with its current operational implementation in the U.S. National Weather Service. EF5 is a framework designed and implemented for real-time flash flood forecasting, thus we choose to keep the name as accurate and descriptive as possible. Regarding the use of the terms "prediction" or "forecasting" versus "simulation", the reviewer is correct in that it would be a different exercise altogether to perfectly replicate a real-time environment by providing EF5 inputs up to a specific time and then launching forecasts into the future. Instead, we provide EF5 continuous rainfall estimates and evaluate the simulation capabilities of the system. As such, we have changed the aforementioned sentence to the following on Pg. 1, Line 6: "The results of the study show that the three uncalibrated water balance models linked to kinematic wave routing are skillful in simulating streamflow." Throghout the remainder of the manuscript, the model outputs that are evaluated are correctly referred to as "simulations".

**Reviewer.** 3. It is not clear that how important adding "Snow (melt) component" in EF5; this seems a newly added feature to EF5 (introduction e.g., P4, L5-7), yet the detail background/examples were not presented in the method. Also, the interpretation of the presented cases (P19, L22-24 linked to the not-used "snow module") needs more solid evidences. What kind of caution (or a priori parameter development as mentioned in P22, L10) should be considered by the users? Please explain more explicitly.

**Response.** This paper is not intended to be an all inclusive review and evaluation of all components available in EF5. Instead, our intention is to describe the framework that was transitioned to the National Weather Service as part of the EF5 initial operational capability. There are additional features of EF5, such as handling snowmelt, assimilation of soil moisture to improve model states, etc., that are under active development and will be fully explained and evaluated in future papers, in synchronicity with their transition to operations. We are not making any claims about handling snow, merely

cautioning users that they should be weary of results when they know frozen precipitation is present. To improve the communication of the intention of the current study, we added the following statement on Pg. 13, Line 15: "The intention of this study is to evaluate the accuracy of the model version that was transitioned to the NWS as part of the EF5 initial operational capability."

**Minor comments**

**Reviewer.** 1. The reference link was broken - Flamig, Z. L., Vergara, H., Clark, III, R., Hong, Y., and Gourley, J. J.: EF5: Version 1.0, doi:10.5281/zenodo.59123, http://dx.doi.org/10.5281/zenodo.59123, 2016.

**Response.** We have updated the DOI referenced to match version 1.2 for this paper.

**Reviewer.** 2. Some acronyms need to be better informed: e.g.,P13, KW, NED, P14 GAMLSS

**Response.** There were numerous acronyms that were not defined upon first use. These have all been fixed. Thanks for catching that.

**Reviewer.** 3. Table1, fix parameters the same as written in P9, IWU has no unit? Check units in other tables as well.

**Response.** Thanks for pointing this out. The IWU parameter is the initial value of soil saturation in percent, used when a soil water content grid (e.g. from a warm-up simulation run) is not available. We have revised the table including now the units, changing the name from "Initial soil water content" to "Initial soil saturation", and including Minimum and Maximum values of 0.0 and 100.0 respectively. We have also changed the nomenclature of the parameters to match that used in the text.

**Reviewer.** 4. P6, 20-22: add reference or provide evidence.

**Response.** The use of the HP solution for diagnosing flash floods and debris flows on burn areas was an outcome through operational use by NWS forecasters in the West. In fact, the developers of EF5 had no intention to transition the HP solution to operations in the NWS. We were only using it to diagnose errors in CREST or in the MRMS rainfall forcings. But, when we took the product off the transition list, forecasters said they wanted to include it. We have thus changed the aforementioned sentence to the following: "Given that the hydrophobic model provides the "worst case scenario" in terms of runoff responses to rainfall, operational forecasters have used it to approximate hydrophobic land surfaces for situations in which the soils are completely saturated, urbanized basins allowing very little infiltration, and for soils that have been affected by wildfire."

**Reviewer.** 5. P20, L1-3, L4-5, L8, P21 L1-2: Need better explanations.

**Response.** The first sentence (now on Pg. 21, Line 1) has been revised to the following: "The results from this study using EF5/CREST, EF5/SAC-SMA, and EF5/HP, all with a-priori, uncalibrated parameters andcoupled to the kinematic wave routing scheme, show no significant systematic errors as a function of watershed scale."

The second sentence (now on Pg. 21, Line 2) has been changed to the following: "It took one week of computer time to simulate streamflow across the CONUS with rainfall estimates being input to the models at a five-min frequency."

We also made the following changes to the sentences being referred to in this comment: "The overall skill of the system is reasonable given the lack of optimized parameters, and on some watersheds the skill is equivalent to that expected with a calibrated hydrologic model. The results in Figure 7 show no significant trend in accuracy as a function of basin area for the range of flash flood basins from 1 km$^2$ to 1,000 km$^2$. The EF5/HP model yields a "worst case scenario" and exhibits large positive bias for most watersheds which is expected behavior for a completely impervious land surface.

We kept the last sentence as-is given this is how forecasters use it in operational practice, as per our response to Comment 4.

**Reviewer.** 6. P22, L2-4: Provide more clear explanation and supporting materials in the results.

**Response.** Sentence has been removed.

**Reviewer.** 7. P22, L15-16, It is not clearly written. Revise the sentence.

**Response.** This sentence has been changed to the following: "As the spatiotemporal resolution of hydrologic models is increasing, the need for validating observations also increases."

---

## Author Response (AR2)

**Response to Topical Editor Decision on "The Ensemble Framework For Flash Flood Forecasting (EF5) v1.2: Description and Case Study" by Zachary L. Flamig et al.**

August 25, 2020

**General comments**

**Reviewer.** Topical Editor Decision: Publish subject to minor revisions (review by editor) (21 Aug 2020) by Jeffrey Neal

Comments to the Author:

Dear Zachary,

Thank you for submitting your revised manuscript to GMD. Having looked at the reviews and your response, coupled to my own reading of the manuscript, I believe this article is suitable for publication in GDM (thank you for adding the model parameters for reproducibility).

In my opinion, the reviewers make two useful suggestions that you only really address properly in the response. Since few will read the response, their suggestions should be captured better in the manuscript. I'm suggesting two very minor edits prior to publication, but please write a brief rebuttal if you think these are not good ideas.

Firstly, reviewer one raises the issue of scale dependence which you agree is a valid point in the response. Given the exchange between authors and the reviewer I would assume this is worth a mention in the literature review or around P4L22 where the scale independence is highlighted?

Secondly both reviewers raise the issue of not including the snow module. I appreciate the reason for this from the response because you are describing what was transitioned, but I would struggle to properly understand that given the new second sentence of section 3.1. The link with the snow module omission (previous sentence) and other module omissions could be more explicit in my view to be more in keeping with the substantial revision suggested by the second reviewer.

Best wishes, Jeff

**Response.**

Dear Jeff,

We appreciate the constructive criticism on the submitted manuscript. Below, we provide our responses and indicate the changes we made to our manuscript. We now believe the peer review process has resulted in an improved product. We are grateful for the time invested by the editors and reviewers in this process.

Regards, JJ

1. Scale issue - We agree more text was needed to discuss the potential scale dependencies of the model simulations. Beginning on p4l22, we added the following text:

EF5 is resolution independent and will work with any DEM resolution having been tested from 0.5 m to 12 km. Note that the a-priori parameters for the water balance modules were derived at 1 km and will need to be resampled to the DEM grid cell resolution. While the overland parameters are linked to observable features of the land surface and soil properties, there can still be a scale dependence of model results due to DEM resolution differences. Finer scale DEMs are capable of resolving more details of the terrain such as steeper slopes in mountainous areas. This can cause the model to produce higher and faster peak flows when going to finer scale DEMs. Furthermore, the routing parameters are scale-dependent and will need to be re-derived for resolutions other than 1 km (additional details provided in section 2.3.2). Links to the parameter grids are provided in the code availability section at the close of the manuscript.

2. Snow module - Agreed. We have now added the following text to begin section 3.1. We believe this communicates the aspects of the initial version that was transitioned to the National Weather Service:

[revised manuscript text omitted]

---

## Author Response (AR3)

**Response to Topical Editor Decision on "The Ensemble Framework For Flash Flood Forecasting (EF5) v1.2: Description and Case Study" by Zachary L. Flamig et al.**

September 2, 2020

**General comments**

**Reviewer.** Topical Editor Decision: Correction needed (review by editor) (28 Aug 2020) by Jeffrey Neal

Comments to the Author: Dear Jonathan,

Thank you for the revised version, I'm happy to accept the scientific content. The executive editor has sent me a reminder about GMD code availability policy, which I need to pass on. Apologies for this late in the day technical correction that I should have spotted earlier.

A GitHub URL is insufficiently persistent in the current version. GitHub themselves tell authors to use Zenodo for this. Could I ask that you follow the instructions here: https://guides.github.com/activities/citable-code/ to archive the exact version of the code which is being presented. Zenodo will provide you with the correct BibTeX entry your bibliography.

Further details on code and data availability requirements are in the GMD model code and data policy: https://www.geoscientific-model-development.net/about/code$_a$nd$_d$ata$_p$olicy.html $https://doi.org/10.5194/gmd-12-2215-2019$.

Any problems please get back to me but I do not envisage needing to review the paper.

Best wishes, Jeff

**Response.**

Dear Jeff,

We appreciate the feedbacks. We have now placed all our code, parameters, and data on Zenodo with DOI certificates and also provide links to Github. We believe this satisfies the data availability policy that was raised by the executive editor. Below is the revised Data and Code Availability Statement that appears at the close of the document:

*Data and code availability.* The source code to EF5 is available on GitHub at https://github.com/HyDROSLab/EF5, on Zenodo at https://zenodo.org/record/569078, has a DOI of 10.5281/zenodo.569078 and is fully documented in Flamig et al. (2017). EF5 is released into the public domain for all use cases. The spatially distributed DEM, routing, and surface water balance parameters as well as potential evapotranspiration forcings are available at https://github.com/HyDROSLab/EF5-US-Parameters, on Zenodo at https://zenodo.org/record/4009759, and has a DOI of 10.5281/zenodo.4009759. Documentation, including the user manual and training videos, can be found at http://ef5.ou.edu. The MRMS radar-based rainfall decadal archive is available at http://edc.occ-data.org/nexrad/mosaic/ with the following DOI: https://doi.org/10.25638/EDC.PRECIP.0001.

Regards, JJ